# Can Large Language Models Distinguish Cause from Effect?

**Zhiheng Lyu**[*†1]    **Zhijing Jin**[*2,3]    **Rada Mihalcea**[4]    **Mrinmaya Sachan**[3]    **Bernhard Schölkopf**[2]

[1]University of Hong Kong, [2]Max Planck Institute, [3]ETH Zürich, [4]University of Michigan
zhiheng.lyu.cs@gmail.com, jinzhi@ethz.ch

## Abstract

Identifying the causal direction between two variables has long been an important but challenging task for causal inference. Existing work proposes to distinguish whether $X \to Y$ or $Y \to X$ by setting up an input-output learning task using the two variables, since causal and anticausal learning have different performances under semi-supervised learning and domain shift. This approach works for many task-specific models trained on the input-output pairs. However, with the rise of general-purpose large language models (LLMs), there are various challenges posed to this previous task-specific learning approach, since continued training of LLMs is less likely to be affordable for university labs, and LLMs are no longer trained on specific input-output pairs. In this work, we propose a new paradigm to distinguish cause from effect using LLMs. Specifically, we conduct post-hoc analysis using natural language prompts that describe different possible causal stories behind the $X, Y$ pairs, and test their zero-shot performance. Through the experiments, we show that the natural language prompts that describe the same causal story as the ground-truth data generating direction achieve the highest zero-shot performance, with 2% margin over anticausal prompts. We highlight that it will be an interesting direction to identify more causal relations using LLMs.[a]

---

[a]Our code and data are at https://github.com/cogito233/llm-bivariate-causal-discovery.

## 1 INTRODUCTION

One of the most impactful deep generative models is autoregressive large language models (LLMs) in natural language processing (NLP), such as the GPT model series [Radford et al., 2018, 2019, Brown et al., 2020], which demonstrate surprisingly strong generalization ability. Autoregressive LLMs are trained to maximize the likelihood $L(\mathcal{D}) = \sum_{i=0}^{N} \log P_{\Theta}(w_i | w_{i-k}, \dots, w_{i-1})$ for an unsupervised large corpus $\mathcal{D} := (w_1, \dots, w_N)$ of $N$ tokens, where $k$ is the size of the context window, and $\Theta$ is the parameters of the generative neural network model.

An important question for deep generative models in general is what they have learned. LLMs have undergone large-scale unsupervised training on almost all available texts, from all public online texts to books and many other sources. It is interesting to check whether the deep generative LLMs have learned causal relations.

In this paper, we look into the causality between two variables, and check whether LLMs can distinguish $X \to Y$ and $Y \to X$ for the causation between two variables. Schölkopf et al. [2012] show that although the two-variable causal relation cannot be easily distinguished by conditional independence tests which are commonly used for causal discovery Spirtes et al. [1993], Pearl [2000], as there are only two variables, it is possible to set up an input-output learning problem to infer the causal direction. There are distinct properties that make causal learning (i.e., models that take the cause as input, and predict the effect) different from anticausal learning (i.e., models that take the effect as input, and predict the cause), such as their different behaviors in semi-supervised learning and under domain shifts. Extending the framework to NLP, Jin et al. [2021] formulate NLP tasks by causal and anticausal learning, which opens the possibility to distinguish $X \to Y$ and $Y \to X$ where at least one variable is text data. Consistent with the independent causal mechanism (ICM) hypothesis, Jin et al. [2021] show that causal learning is more robust against covariate shift, and anticausal learning improves more in semi-supervised learn-

---

[*]Equal contribution.
[†]Work done during summer internship at ETH.

*Accepted for the 38[th] Conference on Uncertainty in Artificial Intelligence* (UAI 2022).

ing, both of which can potentially be used to distinguish cause from effect in language data.

However, the above two types of checks have several conditions that do not apply to LLMs. (1) These checks are conditioned on the fact that the training of the models can be continued, with new unsupervised data or out-of-domain data. However, LLMs are increasingly larger over time, to the extent that university labs cannot easily continue the training given the limited resources in academia. (2) These checks assume the initial causal or anticausal model only sees a task-specific dataset of $X, Y$ pairs from a single distribution. However, the training of LLMs does not use a single task-specific dataset, but, instead, their training is a general-purpose unsupervised learning from massive text data that almost covers all texts that are accessible [Brown et al., 2020, Chowdhery et al., 2022].

Hence, we design a new paradigm to distinguish $X \to Y$ and $Y \to X$ for LLMs. Given the new constraints we are facing when using LLMs, we shift the original input-output learning framework [Schölkopf et al., 2012, Jin et al., 2021] to post-hoc analysis of this deep generative model for text. Specifically, we propose two suggestions: (1) We use zero-shot prompting of LLMs, which can be compared to the commonly adopted hypothesis that modeling along the causal direction is more robust under domain shift. (2) We change the task-specific framing of $P(Y|X)$ and $P(X|Y)$ models to natural language prompts that describe the causal relations of $X \to Y$ and $Y \to X$, respectively. These two adjustments are necessary (although preliminary) attempts to the two-variable causal relation distinction involving LLMs.

Empirically, we conduct experiments with a case study of sentiment classification. We implement different prompts corresponding to the causal relations $X \to Y$ and $Y \to X$, and find that, in general, the LLM zero-shot performance is the best when the prompt is framed in the same causal direction as the ground-truth data. This confirms the effectiveness the proposed approach to distinguish cause from effect with LLMs.

## 2 DISTINGUISHING CAUSE FROM EFFECT IN NLP

Given two random variables $X$ and $Y$ that are known to have direct causal relations of either $X \to Y$ or $Y \to X$, our task is to distinguish which of the two is the true causal relation. In the context of this work that focuses on LLMs, we further assume that the $(x, y)$ pairs come from an NLP dataset, where at least one of the variables is text data.

We use a case study of text classification throughout this paper, which can potentially be extended to other settings or causal graphs involving more variables in future work. Datasets for text classification consist of pairs of text $t$ and its corresponding class label $l$. Suppose that we know that

it is either the text causing the label or the label causing the text, but we do not exactly know which hypothesis is true. To answer this question, this section first introduces the general formulation of causal and anticausal learning (Section 2.1), existing non-LLM approaches to distinguish cause from effect through training (Section 2.2), and our proposed new paradigm to use LLMs for post-hoc analysis to check the causal relations (Section 2.3).

### 2.1 FORMULATION: CAUSAL AND ANTICAUSAL LEARNING

Causal and anticausal learning has been introduced to classify machine learning tasks by whether the task takes the cause as input and predicts the effect, or the effect as input and predicts the cause [Schölkopf et al., 2012]. Formally, given random variables $C$ and $E$, where the former causes the later, namely $C \to E$. Causal learning and anticausal learning aim to learn two opposite functional mappings:

$$\text{Causal Learning: } f : c \mapsto e \qquad (1)$$
$$\text{Anticausal Learning: } g : e \mapsto c . \qquad (2)$$

Recent work extends this formulation to NLP tasks [Jin et al., 2021]. For NLP, in some cases, if we can know how the data was generated, and use these information to know *a priori* what the causal direction is. For example, for the Yelp sentiment classification dataset [Zhang et al., 2015], it is roughly reasonable to assume that the user had some experience with a restaurant, gave a rating, and then wrote the review to justify the rating.

However, there are also cases where it is unknown which the causal direction is between the two variables. For example, many datasets do not clearly indicate the causal direction, or some undecidable cases where even behavioral/cognitive scientists cannot give a conclusion whether it is the behavior that affects the language, or the language affects the behavior. Then, an effective approach to distinguish cause from effect involving language data comes into play.

### 2.2 PREVIOUS WORK: DISTINGUISHING CAUSE FROM EFFECT THROUGH LEARNING

There are several ways to distinguish the cause from effect using the input-output learning setting, due to the different behaviors of causal and anticausal learning. One possible way is to test the improvement brought by semi-supervised learning. Causal learning should not benefit from semi-supervised learning as the marginal distribution $P(C)$ shares no information with the conditional $P(E|C)$ according to the independent causal mechanism (ICM) postulate, whereas anticausal learning may benefit from semi-supervised learning [Schölkopf et al., 2012].

Table 1: Prompts of three causal setups. For the causal direction Review → Rating, there are two possible causal mechanisms, one from the first-person view making a rating based on their own review and the other from a third-person view guessing another user's rating based on their review.

| Causal Setup | Prompt |
|---|---|
| ***Rating → Review:*** 
 (Experience →) Rating → Review | I just finished eating at a restaurant. Then I opened my Yelp app. I first gave a rating in terms of 1 to 5 stars, and then explained why I gave the rating by the following review: `[review text]` The review is an explanation of why I rated it a `[Let GPTs complete]` |
| ***Review → Rating:*** 
 (Experience and) Review → Rating | I just finished eating at a restaurant. Then I opened my Yelp app. I first wrote the following review: `[review text]` Then based on the review, I gave the rating in terms of 1 to 5 stars. I think this restaurant is worth a rating of `[Let GPTs complete]` |
| Review $\xrightarrow{\text{3rdPersonGuess}}$ Rating | I opened my Yelp app, and started to read some reviews of the restaurant that I wanted to try. I saw a user wrote this review: `[review text]` I think this user gave a rating (out of 1 to 5 stars) of `[Let GPTs complete]` |

Another way is to check how the model generalizes to out-of-distribution (OOD) data. Causal learning captures the causal mechanism $P(E|C)$ which is more invariant than the anticausal relation $P(C|E)$, so causal learning should be more robust against domain shifts, as observed in previous studies showing the stronger robustness of causal models against covariate shifts [Jin et al., 2021] and adversarial perturbations [Schott et al., 2019].

However, in the context of LLMs, there are two emerging concerns that hinder us from directly applying the above two approaches.

First, LLMs are increasingly larger, to the extent where it is less and less likely for university labs to continue training the models. As a direct result, the semi-supervised learning approach to distinguish causal and anticausal directions through training is becoming less feasible. For the other approach, although it is also less likely for university labs to finetune LLMs, it is still possible to check OOD generalization by zero-shot or few-shot performance.

Second, there has been an emerging paradigm shift from specific NLP models to general-purpose LLMs that reformat each task to auto-completion. For example, previous NLP models are trained for a specific purpose, such as $P(l|\boldsymbol{t})$ for sentiment classification from text $\boldsymbol{t}$ to label $l$. However, the LLM can potentially reformulate any task as a text completion problem, e.g., "The review '*The food here is great.*' has a rating of `[Let GPTs complete]`." Intuitively, for LLMs, framing a question in natural language makes the best use of the massive free-form text corpus that GPTs are trained on, including all public online texts, lots of books, social media, and so on [Chowdhery et al., 2022].

Based on these two emerging changes that make LLMs different from other machine learning models, we propose

the following paradigm specifically designed for post-hoc analysis of causal relations for the deep generative LLMs.

## 2.3 NEW PROPOSAL: POST-HOC ANALYSIS OF LLMS BY ZERO-SHOT NATURAL LANGUAGE PROMPTS

We improve the previous input-output learning framework to adapt to the new challenges by LLMs. We propose two major adaptations: (1) transforming the learning task using task-specific models of $P(E|C)$ and $P(C|E)$ to LLMs' general-purpose text completion task, and (2) using zero-shot prompting as the main approach to check the robustness against OOD shifts.

**Transforming $P(E|C)$ and $P(C|E)$ to Story Completion.** Since LLMs gradually become general-purpose few-shot learners [Brown et al., 2020], it is important to frame new tasks also in natural language so that it can make good use of the large free-form text corpus on which LLMs are trained. We suggest to shift the traditional task-specific input-output learning models $P(E|C)$ and $P(C|E)$ to the general language-based queries to LLMs. Specifically, we design language-based story completion for each of the three possible causal relations given a dataset of review-sentiment pairs.

In Table 1, we first brainstorm all possible causal setups behind the review-rating pairs in a dataset: (1) (Experience →) Rating → Review: The first possibility is common for naturally generated short user reviews on public websites, where users usually first had some experience, then gave an overall rating, and wrote a review to explain the rationale behind the rating. (2) (Experience and) Review → Rating:

Another possibility could be common for rational decision-makers, or cognitively challenging reviews such as paper reviews, where users first had some experiences, then perhaps wrote down the arguments and evidence to structure the thoughts, and finally gave an *a posteriori* rating based on the reasoning in the review. Here, the review affects (perhaps consciously rationalizes or unconsciously primes) the rating. (3) Review $\xrightarrow{\text{3rdPersonGuess}}$ Rating: Another case of review causing the rating could be the common data annotation setting. For example, Amazon Mechanical Turk (MTurk) workers are shown a piece of text, and asked to provide a class label. We use $\xrightarrow{\text{3rdPersonGuess}}$ to denote that this causal mechanism is by the third person guess (which might introduce additional noises from imperfect theory of mind inferences), but not the first-person cognitive process. The notation helps distinguish from the (2) case where the same user bases their own rating on the review.

Then, for each of the possible causal setups, we compose a corresponding causal story. Our goal is to make the best use of LLMs' seen data on various stories/procedures, and elicit the same causal process through textual descriptions. For example, for Causal Setup 3, we explicitly compose a story from a third-person view "I opened my Yelp app, and started to read some reviews of the restaurant that I wanted to try. I saw a user wrote this review: [review text] I think this user gave a rating (out of 1 to 5 stars) of" and let LLMs complete the remaining text to generate a rating.

**Zero-Shot Prompting.** The other adjustment we propose is to use zero-shot prompting as a substitute for the previous tests against OOD shifts such as training with domain adaptation objectives.

The previous framing of OOD robustness to test causal directions is based on the postulate that the causal learning model should perform better if the model is trained on data from a certain distribution $P_{\text{train}}(C, E)$, but tested on data from a different distribution $P_{\text{test}}(C, E)$, where the two distributions keep a similar causal mechanism $P(E|C)$ but different marginal distributions of the cause, namely $P_{\text{train}}(C)$ and $P_{\text{test}}(C)$.

In the context of LLMs, we align the formulation of zero-shot prompting for LLMs with the above OOD robustness method to distinguish cause from effect. The idea of zero-shot prompting is that given a trained LLM, we directly ask it a question (such as one of the prompts in Table 1), and collect LLMs' answer, which is shown effective in many recent powerful LLMs [Brown et al., 2020, Ouyang et al., 2022, Thoppilan et al., 2022]. Our hypothesis is that, among all the training data that LLMs have seen (including all the online text, etc), there exist all possible causal setups. And the key is that we try to elicit causal or anticausal sub-models from the general-purpose LLMs through the causal stories we compose. For example, it is reasonable to assume

Table 2: Zero-shot performance of prompts corresponding to all three causal setups on the test set of Yelp.

| Setup | Accuracy | Weighted F1 |
|---|---|---|
| Causal Setup 1 | 53.71 | 53.68 |
| Causal Setup 2 | 51.81 | 51.74 |
| Causal Setup 3 | 51.37 | 51.17 |

that when we explicitly ask for the third person view of a review, the LLMs start the $P(l|\boldsymbol{t})$ process by imitating a similar causal process that it saw in its training data.

Based on this, we suggest to directly query LLMs using our causal prompts, and for this dataset of $(\boldsymbol{t}, l)$ pairs with unknown causal direction, the zero-shot prompt with the correct causal direction implicitly asks for $P(E|C)$, and thus will yield the highest performance. Also note that the comparison among the three causal stories are relatively fair because they share the same training data $P_{\text{train}}(C, E)$, and the only difference is which causal process they are querying from the LLMs.

Note that theoretically, we can test the performance of either classifying the sentiment $l$, or generating the review $\boldsymbol{t}$. In real practice, it is much easier to find a metric $d$ to quantify the performance of sentiment classification $d(l, \hat{l})$ than quantifying the performance of review generation $d(\boldsymbol{t}, \hat{\boldsymbol{t}})$, as text generation quality is a notoriously difficult evaluation problem [Celikyilmaz et al., 2020]. Therefore, practically, we fix the task to be sentiment classification $\boldsymbol{t} \mapsto l$, and compare the zero-shot performance of $P(l|\boldsymbol{t})$ and $P(\boldsymbol{t}|l)P(l)$, where $P(l)$ is the prior distribution of labels.

## 3 EXPERIMENTS

**Dataset.** We use the widely used Yelp sentiment classification dataset [Zhang et al., 2015].[1] The data is compiled from Yelp reviews, where a commonly accepted assumption is that the users first select a rating and then write down the review to explain the rating (Causal Setup 1) [Jin et al., 2021]. The original yelp dataset has 650K samples in the training data, and 50K samples in the test data. The dataset has balanced data for each of the five labels corresponding to the $1-5$ rating on Yelp. To make clear the differences between causal and anticausal relations, we use the fine-grained five classes of Yelp review instead of merging them to the coarse-grained positive and negative binary classes.

**Metric.** We report accuracy for all prompts on Yelp, which is the commonly reported metric. In addition, we also report the weighted F1 across the five classes.

---

[1] https://huggingface.co/datasets/yelp_review_full

**Implementation Details.** For the implementation of LLMs, we use the transformers Python library [Wolf et al., 2019]. Since we do not have enough computational resources for GPT2-xl, we use the second largest GPT2 model, GPT2-large, for our experiment, which is the best autoregressive LLM that we can run and also fits the task (which is more suitable than T5 for this free-form text completion setup). In future work, we will explore more variants of GPTs. For computation efficiency (to save inference time of LLMs), we use some random subsets of the training set, 10K samples to calculate the prior distribution, and another 10K to select the best prompt among a large set of possible paraphrases.

**Main Results.** Our main experimental results are in Table 2, where we can see that the true causal prompts (Causal Setup 1) show higher performance than the other ones (Causal Setup 2 & 3) by a clear margin of 1.9% by accuracy and 1.94% by weighted F1. For future work, it will also be interesting to analyze GPTs of different sizes and see if there is a scaling effect of clearer causal distinction as the models get larger.

## 4   FUTURE WORK

One natural direction of future work is to add more experiments and analyses: We plan to extend from the Yelp dataset to more various datasets with different causal nature. As for analysis, it will be good to include more different sizes of GPTs, including InstructGPT, GPT3, GPT2-xl, GPT2-medium, GPT2-small, and distilGPT to see if there is a scaling effect such as the larger the model, the more distinct advantage the true causal setup will show. It will also be interesting to run text adversarial attack algorithms to check the robustness of LLMs under different causal prompts.

In addition, this work could pave the way to many other interesting directions for future work: (1) In the scope of this paper, we introduce a paradigm to use LLMs to distinguish the two-variable direct causal relation, namely $X \rightarrow Y$ or $Y \rightarrow X$. For future work, this framework could be potentially extended to more complicated causal relations involving more variables.

(2) The current proposal suggests a path to check langauge-related causal relations, which could be of interest to interdisciplinary researchers, such as cognitive scientists and behavioral scientists, since it is an important research direction to distinguish how language affects actions and thinking. There could also be more analysis done by comparing LLMs outputs with surveys from human subjects, such as comparing LLMs responses to Causal Setup 3 with an actual setting of asking MTurks to judge Yelp reviews.

(3) Extending the input-output learning framework for LLMs is not the only way to extract causal relations from the deep generative LLMs. Another line of research in NLP could also form it as question answering, such as asking LLMs directly about causal facts of the world, e.g., what causes an object to move at a constant velocity, to check whether LLMs will answer nothing (as in Newton's law of physics), or a constant force (as in Aristotle's words).

## 5   CONCLUSION

This work addresses an interesting problem: how can we distinct cause from effect using LLMs? We derive inspirations from previous work that formulates this question by checking the performance differences of the input-output learning problems, namely causal and anticausal learning. We propose a novel paradigm to adapt such ideas for LLMs, by suggesting to transform the task-specific input-output learning to general natural language prompts with a causal story and replace the training-based differentiation of cause and effect pairs with zero-shot prompting of LLMs. This work opens a new direction to connect traditional cause and effect distinctions with the emerging trend of LLMs, and paves the way for various possible future work on post-hoc analysis of causal relations extracted from LLMs.

### Author Contributions

Zhiheng Lyu conducted all the experiments and analyses.

Zhijing Jin designed the project and wrote the paper.

Rada Mihalcea, Mrinmaya Sachan and Bernhard Schölkopf supervised the project.

### Acknowledgements

This material is based in part upon works supported by the German Federal Ministry of Education and Research (BMBF): Tübingen AI Center, FKZ: 01IS18039B; by the Machine Learning Cluster of Excellence, EXC number 2064/1 – Project number 390727645; by the Precision Health Initiative at the University of Michigan; by the John Templeton Foundation (grant #61156); by a Responsible AI grant by the Haslerstiftung; and an ETH Grant (ETH-19 21-1).

Zhijing Jin is supported by PhD fellowships from the Future of Life Institute and Open Philanthropy. We also thank OpenAI for granting Zhijing and the team free access to OpenAI's API through the Researcher Access Program.

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
