# OpenReview forum: "Can Large Language Models Distinguish Cause from Effect?"
_auai.org/UAI/2022/Workshop/CRL — CRL@UAI 2022 Poster_

### Official Review · Reviewer_cipb · 2022-06-22
**Clear and well written, sentence completion enables asymmetric predictions in NLP. A biased training may lead to mispredicted causal effects, insufficient experiments.**

**Rating:** 3
**Confidence:** 3

**Review:**

## Summary
This paper focuses on causal relationships in natural language processing where at least one of the observables is a text sentece. Specifically, the problem of getting the causal direction in the bivariate case is addressed. The causal direction is estimated from a pretrained Large Language Model (LLM) using causally-aware zero-shot prompts. The framework is instantiated on sentiment classification.

## Strengths
* The problem this work addresses is of high interest since natural language has a high representational capacity. Potentially, we can cast classic causal problems as natural language and solve them using NLP.
* Motivations underlying the work are explicit and well written: LLMs are prohibitive to train and general purpose.
* The contributions are stated in a concise and clear manner: causally-aware prompts and zero-shot post-hoc probing
* In contrast to previous work addressing causality in settings where input and output are symmetric, e.g., machine translation, prompting with sentence completion enables to use a probing module for highly asymmetric tasks, e.g., sentiment analysis

## Weaknesses
* Zero-shot prompts and learning the observables: understanding the causal relationship between two variables requires learning the two observables in the first place. How do we enforce this on a pretrained LLM? How can we set cases where the model has not learned the variables apart from cases where it has? For instance, can we predict the relationship between PKA and JnK for Protein-signaling as in Sachs et al.?
* Zero-shot as OOD?: zero-shot in the context of NLP refers to evaluating a model on a task it has not be explicitly trained for, e.g., the model is trained as denoising AE but evaluation asks for sentiment analysis. OOD robustness estimates the ability of the model to cope with changed causal mechanism instead. Zero-shot does not imply a mechanism shift. Consider training the model as denoising AE on 3 observables X, Y, Z. Evaluating the prediction of Y from X does not imply that the functional mapping f: X -> Y has changed.
* Missing knowledge of model training strategy does not disambiguate causal relation: Jin et al. leverage the known training setting of the model to disambiguate the causal direction. A model trained in a fixed direction is assessed to understand whether the learning is aligned with the ground truth causal direction. Consider a model trained to predict Y from X:
	* If it is causal i.e., easier to adapt on OOD data and no gains with SSL, the learning is aligned with the underlying causal relationship, hence, we can conclude that X -> Y.
	* If it is anticausal, it is learning to predict the cause from the effect. i.e., Y -> X
	The reasoning does not hold without knowledge about the training of the probed model as in LLMs. The training may bias the prediction towards one causal direction.
	Suppose a LLM is trained to reconstruct on 5% setup 1, 90% setup 2, 5% setup 3 for restaurant reviews. I would expect the model to perform best on the second causal setup, independently of the data causal direction for cafe' reviews.
* A More extensive evaluation is needed to support the claim. Authors should include multiple tasks and multiple models to back up their hypothesis.

### Questions
* The work is closer in spirit to Jin et al.'s observation that there exists a large performance gap between causal and anticausal learning. Learning the direct causal mechanism is easier than the inverse one. In contrast to OOD robustness, model generalization may act as heuristic for cause-effect prediction. The rule of thumb holds under the assumption of fairness  between causal and anticausal predictions. Though, as before, the LLM training may bias the performances (see weaknesses). Moreover, I would expect the advantage of the causal model to vanish as we see more data, Bengio et al. 2019
* Do we really need a large language model? Can we use a smaller model as a probing method to evaluate the direction of learning?

---
Bengio, Yoshua, et al. "A Meta-Transfer Objective for Learning to Disentangle Causal Mechanisms." _International Conference on Learning Representations_. 2019.

K. Sachs, O. Perez, D. Pe'er, D. A. Lauffenburger and G. P. Nolan. Causal Protein-Signaling Networks Derived from Multiparameter Single-Cell Data. Science, 308:523-529, 2005.

Jin, Zhijing, et al. "Causal Direction of Data Collection Matters: Implications of Causal and Anticausal Learning for NLP." _Proceedings of the 2021 Conference on Empirical Methods in Natural Language Processing_. 2021.

---

### Official Review · Reviewer_dzgY · 2022-06-27
**A human-like approach to causal discovery**

**Rating:** 5
**Confidence:** 3

**Review:**

**Summary**: This paper uses large language models (LLMs) to perform causal discovery between two variables (one in text form) without unobserved confounding. The approach trains two models, one in a causal direction and the other in an anti-causal direction, and then evaluates their performance in a zero-shot learning environment, where the model trained in the causal direction is expected to perform better. To take advantage of the large amount of training data in text form, the model is designed to provide outputs in the form of auto-completing text.

**Review**: The idea of using text auto-completion for causal discovery is certainly an interesting and unique idea. The prompts provided in Table 1 resemble a very human-like approach to determining causal direction. Still, the paper appears to lack concrete contributions. On the theory side, I am concerned that the approach is not sound. In general, causal direction between two variables cannot be determined from observational data alone. Doing so requires additional assumptions, and it is not clear what assumptions are used in this paper or whether those assumptions would allow for a correct inference. It seems that the approach depends on the availability of data in another domain where only the distribution of the cause is changed, so some analysis on this approach would be helpful. The lack of unobserved confounding is also not discussed much, but it is unclear if this is a reasonable assumption in NLP settings, since this may be the reason why causal direction is difficult to determine in many settings. On the experimental side, the results seem to be rather minimal. It is understandably difficult to run many experiments using large language models due to resource limitations, but it would have been interesting to see more analysis on the trained models in the existing experiments, such as examples of outputs in the train and test domains. Overall, although the idea is interesting, I believe that it needs to be explored a little further both theoretically and empirically.

---

### Official Review · Reviewer_KppY · 2022-06-28
**Comments for the authors**

**Rating:** 7
**Confidence:** 4

**Review:**

Summary:
This paper considers the problem of discovering causal relationships between two variables, X and Y, when one of the variables is text. The paper follows previous observations in this setting that if text (X) --> label (Y), then a supervised model trained to predict Y from X will have better out-of-distribution prediction performance than if label --> text. The paper implements this observation for causal discovery in the new setting where we use large language models (LLMs). The challenge in the LLM setting is that P(Y|X) models can't be fine-tuned easily. The paper proposes to replace prediction with natural language prompts and zero-shot inference tasks. They introduce new prompts and zero-shot tasks that aim to estimate whether X --> Y or Y ---> X. They empirical evaluate their method with texts and labels from a text --> label process. The results provide mild evidence that the method can tell apart causal directions.

Strengths:
+ The writing is very clear. The authors introduced the setting and background well, discussed their contributions clearly and explained their method and empirical studies well.
+ Showing new ways of replacing fine tuning with natural language completion tasks and zero-shot tasks is clearly a growing area of interest and is of interest to the community.

Some concerns and comments for the authors:
+ In section 2.1, the authors note that the motivating problem is that in the context of NLP, "there are [...] cases where it is unknown which the causal direction is between the two variables." It would useful to give more concrete examples of when, in language settings, we don't have a priori knowledge about whether text --> label or label --> text. It would also be good to see these examples and this motivating question posed in the intro as a clear problem that the paper seeks to address. Right now, there's a bit of ambiguity as to why the paper wants to tackle this causal discovery problem in the context of NLP. A clearer set of motivating questions or goals would help.

+ I could use more technical justification and reasoning as to why the prompts need to encode something about the posited causal structure. As a contrast, when we train P(E|C) or P(C|E) models using standard text prediction methods, the model has no awareness about the posited causal structure so it isn't entirely clear to me why the prompting version same task would require this extra context about a causal structure guess that we're trying to evaluate. From the zero-shot prompting section, I gather that the 'causal story' part of the prompt serves to filter the training data the model had seen somehow, guiding it to do the task based on the samples that adhered to that causal story?

+ Is there any way to obtain standard errors in zero-shot settings? it might be helpful to understand if the empirical performance differences are notable or not.

+ As a related question, is it well-studied or known that by including a story in the prompt about what kind of causal structure is being posited, we can guide LLMs to perform completion based on a relevant subset of training samples? Perhaps, the empirical results are not yet super conclusive because the little causal story at the beginning of the prompt doesn't push the model enough to use different information to answer each of the differing prompts.

---

### Meta-Review · Program_Chairs · 2022-07-05

**Recommendation:** Accept (Poster)
**Confidence:** 4

**Metareview:**

While there was quite some disagreement for this paper and some of the reviewers had some concerns (especially reviewer cipb and partially also reviewer dzgY), we feel that this paper would still provide a very interesting discussion in the UAI CRL workshop, so we recommend acceptance. We do encourage the authors to try to address some of these concerns and potential misunderstandings in an updated version.

---

### Decision · Program_Chairs · 2022-07-06

Accept (Poster)